# A Universal Malicious Documents Static Detection Framework Based on Feature Generalization

**Xiaofeng Lu** [1,*] , **Fei Wang** [1] **, Cheng Jiang** [1] **and Pietro Lio** [2]

1   School of Cyberspace Security, Beijing University of Posts and Telecommunications, Beijing 100876, China; wf751620780@gmail.com (F.W.); JC@bupt.edu.cn (C.J.)
2   Computer Laboratory, University of Cambridge, Cambridge CB3 0FD, UK; PL219@cam.ac.uk
*   Correspondence: Luxf@bupt.edu.cn

**Abstract:** In this study, Portable Document Format (PDF), Word, Excel, Rich Test format (RTF) and image documents are taken as the research objects to study a static and fast method by which to detect malicious documents. Malicious PDF and Word document features are abstracted and extended, which can be used to detect other types of documents. A universal static detection framework for malicious documents based on feature generalization is then proposed. The generalized features include specification check errors, the structure path, code keywords, and the number of objects. The proposed method is verified on two datasets, and is compared with Kaspersky, NOD32, and McAfee antivirus software. The experimental results demonstrate that the proposed method achieves good performance in terms of the detection accuracy, runtime, and scalability. The average F1-score of all types of documents is found to be 0.99, and the average detection time of a document is 0.5926 s, which is at the same level as the compared antivirus software.

**Keywords:** malicious document detection; static detection; feature generalization; machine learning

## 1. Introduction

Documents are efficient, convenient, and safe information carriers. However, via some deliberate designs, an attacker can run malicious code on a document, or can use the vulnerabilities of document parsing software to invade the operating system; these documents are called malicious documents. According to statistics, malicious documents are mainly Portable Document Format (PDF), Word, Excel, Rich Test format (RTF), PowerPoint, Graphics Interchange Format (GIF), Joint Photographic Experts Group (JPEG), and Portable Network Graphic Format (PNG) documents. Among them, PDF and Word documents respectively account for 60.1% and 38.7% of malicious documents [1], which is mainly because these documents are more widely used for information transmission. Moreover, PDF and Word documents can contain almost any type of document as the document content, and script programs can be inserted to increase the specific functions of the document.

In January 2014, an attacker launched an attack by disguising a malicious PDF document as a document of the Israeli Ministry of Defense, and then sending the PDF as an attachment to an email [2]. In some kinds of malicious PDF attacks, the PDF reader itself contains a vulnerability or flaw that allows a file to execute malicious code [3]. Many attacks try to abuse this flaw via the use of social engineering or by hosting malicious PDF files on the Internet. Attackers can induce people to open malicious documents by sending emails with malicious attachments. Just opening the PDF file could exploit a vulnerability [4]. In December 2018, a threat report from the security company MalwareBytes warned that the banking Trojan/download/botnet and its common complicity, Trickbot, mainly use email to distribute malicious Office documents that use PowerShell to download malware [5]. Malicious document attacks not only bring huge risks to individuals, but also seriously threaten the security of enterprises [6].

To date, there has been some research on the detection of malicious documents. This research includes the detection of malicious code based on image processing using deep learning [7], the detection of malicious documents for business process management based on a multi-layer abstract model [8], the detection of unknown malicious Microsoft Office documents via the use of designated active learning methods based on a new structural feature extraction methodology [9], the detection of malicious PDF documents based on mixed features [10], and the improvement of the detection of malicious Office documents using one-side classifiers [11].

Currently, the research on malicious document detection basically only involves one certain type of malicious document, and related research has rarely investigated whether the detection features proposed for one type of document can be used to detect other types of documents. Because the types of documents are rich and diverse, it is inefficient to study the detection features and design classifiers for each type of document. For this reason, in the present study, the common features of malicious PDF and Word documents are extracted to detect multiple types of malicious documents, and this technology is defined as feature generalization.

A universal static detection framework for malicious documents based on feature generalization is developed in this research. The framework uses four feature dimensions, namely specification check errors, the structure path of the document, document code analysis, and the number of objects. By combining these four types of features, the decision tree and random forest models are used for classification. Through experimental comparisons with antivirus software and other research, the effectiveness of the proposed universal static detection framework for malicious documents based on feature generalization is proven.

The main contributions of this work are as follows:

1.  In this research, document specification check errors are innovatively utilized as a component of feature engineering;
2.  A feature generalization method, which extends the features of the structural path dimension used in PDFs to other types of documents, is proposed. The application of various software to obtain various features of documents is explained in detail;
3.  The proposed method is verified on two datasets, and is compared with Kaspersky, NOD32, and McAfee antivirus software. The experimental results demonstrate that the proposed method achieves superior performance.

The remainder of this paper is structured as follows. Section 2 discusses related work. Section 3 introduces attack forms of malicious documents. Section 4 presents the feature engineering based on feature generalization, including specification check errors, the structure path of the document, document code analysis, and the number of objects. Section 5 introduces the static detection framework. Section 6 presents our experiment and result. Section 7 discusses the shortcoming of our method. Section 8 concludes the paper and discusses the future work.

## 2. Related Work

### 2.1. Static Analysis Method

Malicious document detection methods are mainly divided into static analysis and dynamic analysis. The main idea of dynamic analysis is to perform security analysis by using virtual machines to execute suspicious targets [12–15]. Compared with static analysis methods, dynamic analysis has lower accuracy and lower detection efficiency.

Early detection of malicious PDF documents mainly focused on JavaScript [16–18]. The detection system PDFRate was based on content metadata [19]. Later, Srndic and Laskov designed a malicious PDF detection method based on structured path [20]. The structural path is used as the detection feature to deal with unknown security threats.

Nissim et al. [21,22] proposed using active learning, which performs heuristic teaching detection and usability detection on PDF documents. They built a structure-based detection model, and performed detection and relearning. The main advantage of the model was the

reduction of the workload of manual analysis of PDF documents, but the detection process was more complicated and the efficiency was low.

Young-Seob Jeong et al. [23] designed a convolutional neural network (CNN), which takes a byte sequence of an inexecutable stream as input and predicts whether the input sequence contains malicious operations. However, only the PDF was tested and the training time of the CNN model largely depends on the performance of the GPU (Graphic Processing Unit). Du Xuehui et al. [10] selected seven important features as universal features and used a random forest algorithm to detect malicious PDFs. These features were extracted through the structural feature extraction algorithm.

### 2.2. Features of Static Analysis

At present, most researches extract features from structural paths or code keywords and perform malicious document detection. Benign samples are usually generated by documentation tools and have high compliance. Attackers usually deliberately design the content of the document or embed malicious code, which will make the structure of the document different from a benign document [24].

Many researches focus on the detection of malicious PDF documents, such as the research of Nissim et al. [24]. We refer to the PDF detection method and found that structural paths or code keywords can achieve good results in the selection of document features.

Keywords are the features of the code embedded in the document. Malicious samples of VBA (Visual Basic for Applications) programs embedded in Word documents usually use keywords such as "URLDownloadToFile" and "AutoExec". The former is used to download bits from the Internet and save them in a document. The latter is used to automatically execute batch commands. These two keywords rarely appear in ordinary samples, but they usually appear in malicious samples. Therefore, the VBA code can be used to detect the maliciousness of Word documents.

PDF documents allow JavaScript code to be embedded in. Some PDF readers contain some vulnerabilities or flaws that allow a file to execute malicious code [3]. The keywords in the JavaScript code and the keywords in the PDF document are used in the malicious PDF detection, and good results have been achieved [25,26].

Li and Shafiq et al. proposed the "format agnostic theory" to detect malicious document content [27]. In recent years, the detection of malicious PDF documents has achieved better results by document structure. Srndic and Laskov proposed the concept of a structured path that had a good detection effect and a strong ability to respond to unknown security threats. Maiorca et al. [25] proposed a detection scheme based on structure and content. Chen et al. [28] extracted the structural path features, and the input features have 3514 dimensions; however, its robustness is weak. Maiorca et al. [18] proposed a bypass method based on structure detection for this scheme, which is called "reverse mimicry", and it proved the feasibility and simplicity of this attack.

Some researchers applied the structure path of the PDF to the docx document. Nissim et al. [9] trained the xml tag of the docx based on Open-xml and the document path of the compressed package as the structure feature of the docx document, and this method obtained a high detection rate. Lu et al. [29] proposed a malicious word document detection method based on multi-view features learning. They analyzed Word documents from four independent views: VBA functional words, Ole file object formats, structure paths, and specification errors. The method of analysing malicious documents from independent different views is helpful.

Generally, the detection methods based on one-dimensional features are easily bypassed. The content of the malicious document or embed malicious code makes the structure of the document, specification compliance, structural order, number of objects, and malicious code inserted in the document different from a benign document. In this paper, we will combine multiple features to improve detection performance.

## 3. Attack Forms of Malicious Documents

Attackers can use documents to launch attacks in a variety of ways. Some documents can contain specific scripts and execute external commands, and some can use document parsing vulnerabilities on websites to launch web attacks. This brings great difficulties for security personnel to analyze document security. Therefore, understanding the document structure and the common attack methods by which attackers exploit documents can help defenders more effectively diagnose the security of documents. In this section, the main attack forms of PDF and Office documents are introduced.

### 3.1. Main Attack Forms of PDF Documents

PDF documents allow for the embedding of JavaScript code. JavaScript code is usually used for form filling and office automation, while embedded documents can present a richer content display for PDFs. JavaScript code attack and embedded file attack are common forms of PDF attack [30–32].

#### 3.1.1. JavaScript Code Attack

Some PDF readers contain some vulnerabilities or flaws that allow a file to execute malicious code. The vulnerabilities exist in the character parser and JavaScript engine. The contents of PDF documents are stored in stream objects, which generally utilize compression technology to reduce the file size. During the parsing process, the relevant dynamic library will be called. During this process, some functions might be not secure (for example, if there is an overflow vulnerability), which will become the source of attacks. Attackers can exploit the related vulnerabilities and JavaScript codes of PDF readers, and can use heap-spraying technology to achieve attacks [30].

#### 3.1.2. Embedded File Attack

Some types of documents, such as executable file (EXE), Small Web Format (SWF), Flash, and other file types, can be embedded in PDF documents [31]. These types of files may also have some vulnerabilities that can be exploited by attackers. For example, attackers can use Flash vulnerabilities to launch attacks [32].

### 3.2. Main Attack Forms of Office Documents

According to an analysis of the Word samples used in this study with Visual Basic for Applications (VBA) codes, the proportion of malicious documents was found to be 98.13%. The malicious samples without VBA codes were analyzed, and it was found that the causes of malicious behavior were malicious images (JPEG, GIF, PNG), malicious Object Linking and Embedding (OLE) objects, remote requests (Uniform Resource Locator (URL), domain name, Internet Protocol (IP) address), Dynamic Data Exchange (DDE) commands, Common Vulnerabilities and Exposures (CVE) exploits, and other reasons.

According to the literatures, we summarize five important attack methods: embedded VBA malicious code [33], embedded OLE objects [9], program vulnerabilities caused by incomplete detection based on specifications [34], DDE [35], and the malicious images, malicious flash and other multimedia documents inserted in the body of the document [36].

#### 3.2.1. VBA Malicious Code

VBA is a high-level macro language. Word documents embedded with malicious VBA code usually automatically run VBA programs at startup, perform remote download tasks, modify the registry, and destroy important file data on the computer [33]. The function words appearing in the VBA code can imply the possible functions of the program; thus, extracting the function words from the VBA code can help analyze the security of the document.

### 3.2.2. OLE Objects

OLE is an object-oriented technology that users can use to develop reusable components. OLE objects embedded in malicious documents are usually executable files, registries, and other files. The executable files include Portable Executable (PE) files, OLE Control Extension (OCX) libraries, JAVA Archives (JAR) packages, 16-bit Disk Operating System (DOS) files, and Link (LNK) files. Command-line code can implement remote download commands or start external processes to do more things or implement file encryption ransomware. For example, CVE-2018-8174, also known as "Double Kill," is a member of a family of exploits that leverages the OLE functionality of Microsoft Office to download a web page containing a custom VBScript and immediately run it [37].

### 3.2.3. Document Specification Vulnerabilities

Normally, the document reader will perform a specification check according to the items specified in the document specification, while ignoring the inspection of other data parts. This is mainly because a too-thorough specification check will reduce the operating efficiency, and normal documents also include irregular data. The strategy of ignoring errors and performing simple specification checks gives attackers an opportunity to attack. Therefore, attackers use document readers to parse the documents, search for vulnerabilities, design the contents of the documents, and ultimately achieve the purpose of the attack. Figure 1 presents a case of a malicious DOCX document using the CVE-2017-0199 vulnerability. The vulnerability is that the document reader does not check the word/_rels/webSettings.xml.rels file, and attackers exploit this vulnerability to download files from a remote server [38].

```xml
<?xml version="1.0" encoding="UTF-8" standalone="yes"?>
<Relationships
    xmlns="http://schemas.openxmlformats.org/package/2006/relationships">
    <Relationship Id="rId1" Type="http://schemas.openxmlformats.org/
    officeDocument/2006/relationships/frame" Target="http://78.128.92.108/
    Document/word.doc" TargetMode="External"/>
</Relationships>
```

**Figure 1.** Example content of word/_rels/webSettings.xml.rels.

### 3.2.4. DDE Commands

In October 2017, SensePost released an article that explained how it is possible for arbitrary code to be executed from a Microsoft Word document without using any macros or scripts [39]; this technique is called dynamic data exchange (DDE). This attack is very effective, and is therefore widely used in malware campaigns and red team assessments [35]. DDE is a legitimate Microsoft Office functionality, and the DDE protocol is a set of messages and guidelines. Malicious Word documents usually include the insertion of DDE commands, which will pop up a dialog box when the program starts to ask the user whether to execute them.

### 3.2.5. Pictures and Other Media Files

Some Word documents are inserted malicious images or flash files, which cause the document reader to run abnormally, so that Shellcodes can be executed [40]. Advanced Persistent Threat (APT) groups widely exploit the new Adobe flash 0-day vulnerability that is inserted into Microsoft (MS) word documents. Attackers can execute the malicious flash object via MS word documents into the victims machine [41].

### 3.3. Malicious Forms of Image Documents

As one of the most common forms of information exchange, tens of thousands of images are generated every day. Images are often embedded in ordinary documents to convey more information. The most commonly used image formats on the Internet are GIF,

PNG, and JPEG. GIF is widely used for web browsing, while JPEG and PNG are not only used for web browsing, but most often used in the main body of the document content.

The malicious behaviors of images mainly include the following aspects: forge file headers [42], CVE vulnerabilities [43], inserted codes [44], embedding HTML codes, and hiding malicious data [45]. More information about malicious forms of image is included in the Appendix A.

### 3.4. Malicious Forms of RTF Documents

RTF (Rich Text Format) is a rich text format document, which is similar to a DOC format (Word document), but the content of the document can be opened and edited with notepad like a PDF document. An RTF document is composed of a file header and a file body. Both are composed of text, control words, control characters and groups. The control character is composed of "\" and non-alphanumeric characters, and the control word is composed of "\letter control sequence <delimiter>", which is a special format command. The group is surrounded by curly braces ({}), and there are texts, control words or control symbols in curly braces. The information in each group describes the text and text attributes that it modifies.

RTF attacks mainly include array overflow vulnerabilities [46] and OLE object vulnerabilities [47]. More information about malicious forms of RTF documents is included in the Appendix B.

### 4. Feature Engineering

For this study, 290,542 files from Contagio, VirusTotal, and VirusShare were collected, and the benign samples were mainly sourced from Baidu Library and academic sharing for academic purposes [48]. Through the analysis of the 290,542 files, three kinds of malicious document attacks were identified, namely (1) using the code execution function of the document itself to execute the malicious code, (2) using the vulnerability of the document reading software to run the shellcode when parsing the document, and (3) realizing the embedding of malicious documents via the extended content of the document (such as OLE objects). This provides a solution for feature engineering for document security detection.

### 4.1. Specification Check Error

Each type of document has its own document specification standard. The formulation of the document specification standard enables the document to be opened and read by more software, and guarantees the security of the document.

On the one hand, when opening a document for parsing, ordinary software or document readers usually do not perform integrity checks on the document in accordance with the document specification standards. This is because if a specification check is conducted, the document analysis time will be increased and the software operation efficiency will be reduced. The common method is to parse the document, capture abnormal behavior of the software during document analysis, and report document corruption only when a fatal error occurs. When there is a small error in the document, to give the user a good experience, the software usually ignores the erroneous document areas and continues to parse the document.

On the other hand, it is not safe to directly use software, such as the reader, to directly parse the document, and this operation is a form of dynamic operation. For malicious documents, the direct use of software to parse and wait for errors to be reported is equivalent to executing malicious documents without any protection. Therefore, it is not safe to use the document parsing function of the reader, whereas the performance of a static specification check on the sample is a very safe method.

### 4.1.1. Office Document Error

The OffVis tool proposed by Microsoft can complete the specification inspection of Office documents. It can detect a small number of publicly exploited vulnerabilities when reading documents.

Because the OffVis tool does not have a command operation version, and because the code of the project is written in Python, the OffVis tool is processed as follows:

1. The open-source tool ILSpy is used to decompile cases.dll and GUT Architecture.dll;
2. The code is modified to retain the parsing module and the XML serialization interface of the object;
3. The functions in it are called through the Python CLR module.

Although the OffVis tool can logically judge the data in the specified data field for some CVE vulnerabilities, the sample analysis revealed that some benign documents still have CVE vulnerability detection reports. This indicates that the CVE vulnerability report cannot be used as the error message of the document to participate in the feature selection.

Table 1 presents the proportion distribution of the files verified by OffVis specification in MicroSoft DOC documents among the malicious and benign samples. It can be seen from the table that a large number of samples in malicious DOC documents have specification errors, while most benign DOC samples do not have detected specification errors. Therefore, specification errors can be considered as a candidate feature.

**Table 1.** The proportion distribution of normative inspections of MS-DOC documents.

| DOC | No Errors | Specification Errors |
|---|---|---|
| Malicious | 5.67% | 94.33% |
| Benign | 67.69% | 32.31% |

### 4.1.2. PDF Document Error

PeePDF is an open-source tool written in Python for the analysis of the security of PDF documents. For PDF document detection, the PeePDF tool can first be used to complete the analysis of the PDF document, from which document error information, object information, object flow, and other information can be obtained. Then, relevant features can be extracted from this information.

The PeePDF tool performs non-mandatory scanning. If the tool considers PDFs to be suspicious, it means that they may contain suspicious elements, such as code execution, format errors, incorrect X-ref tables, and corrupted titles. It is worth noting that these elements may also be present in legal samples. Therefore, PeePDF cannot be used as a malicious detector because it generates too many false positives.

The following four types of error-related information were extracted as candidate features:

1. Errors in each part: the origin of the error information is three-fold, namely (1) the error of the document object, (2) the error of the X-ref table, and (3) the location and information of the exception caused by the PeePDF tool when parsing the PDF document;
2. The number of error objects;
3. Keywords that do not meet the PDF specification; and
4. Code with known vulnerabilities.

### 4.1.3. Image Specification Error

The types of images studied include GIF, JPEG, and PNG images. Because the structure of the image document is relatively simple, an image analysis tool is manually written in the detection system to analyze the image content and object information. Moreover, the wrong data and data outside the document specification during the parsing process are recorded. When data are encountered outside the document specification, this portion of the data will be analyzed.

According to statistics, the types of data outside the document specification mainly include the following: PE documents, import tables of PE documents, iframe tags, HTML codes, JavaScript codes, image documents, JAR packages, compressed packages (Roshal Archive (RAR), Phil Katz's Compressed File (ZIP), 7-Zip (7Z)), Executable and Linking Format (ELF) documents, and PDF documents. Table 2 presents the distribution of the specification check results of GIF documents, which reveals that the vast majority of malicious GIF documents do not meet the GIF document specification, whereas most of the benign samples meet the GIF document specification.

**Table 2.** The proportion distribution of normative inspections of GIF documents.

| GIF | No Errors | Specification Errors |
|---|---|---|
| Malicious | 5.31% | 94.69% |
| Benign | 98.93% | 1.07% |

### 4.2. Structure Path

In recent years, scholars have applied structured paths to PDF documents, and the experimental results demonstrate that structured paths can be effectively used as the malicious features of classified documents; however, such features also have certain limitations, i.e., they can easily be used to avoid detection by attackers [18].

In this study, a new structure path extraction scheme is proposed, namely that the sequence of objects appearing in the document is taken as the structure path.

The object structure of a benign document is not intentionally constructed, whereas the document structure of a malicious document will be deliberately disguised to trigger loopholes and deceive users about the object structure. Therefore, there are many differences in the order of the object structure.

When directly obtaining the structure path of the object in the document, the path explosion problem will occur due to the excessive length of the path; thus, the structure path of the object should be truncated by the n-gram method.

#### 4.2.1. Office Document Structure Path

In this section, the structure path of a Word document is taken as an example to illustrate the process of structure path extraction. For DOC documents, as reported in Section 4.1, the OffVis tool is used to obtain the result of serializing objects generated after DOC document parsing into XML. The DOCX document itself is a compressed ZIP document based on XML. When the DOC document object is serialized into XML, it can be processed uniformly like a DOCX document.

To obtain the structure paths of DOC and DOCX documents, the path of the tags must be obtained from the XML document, and for other non-XML documents, such as DOCX documents, the relative path of the document is used as the structure path. For example, the VBA program in the DOCX document will be packaged by using the vbaProject.bin document, and an image document will be placed under the "word\media" path and denoted as imagexxx.jpg. Thus, the paths of these documents are recorded, and the tag of each XML document is recorded as the path. For example, Figure 2 presents the content of a sample of "word\webSettings.xml." According to the tags in the XML document, eight structure paths will be generated:

word\webSettings.xml-w:webSettings-w:frameset-w:framesetSplitbar-w:noBorder
word\webSettings.xml-w:webSettings-w:allowPNG
word\webSettings.xml-w:webSettings-w:frameset-w:frameset-w:frame-w:linkedToFile
word\webSettings.xml-w:webSettings-w:frameset-w:frameset-w:frame-w:sourceFileName
word\webSettings.xml-w:webSettings-w:optimizeForBrowser
word\webSettings.xml-w:webSettings-w:frameset-w:framesetSplitbar-w:color
word\webSettings.xml-w:webSettings-w:frameset-w:frameset-w:frame-w:name
word\webSettings.xml-w:webSettings-w:frameset-w:framesetSplitbar-w:w

```
1   <?xml version="1.0" encoding="UTF-8" STANDALONE="yes"?>
2   <w:webSettings xmlns:mc="http://schemas.openxmlformats.org/markup-compatibility/2006"
3       xmlns:r="http://schemas.openxmlformats.org/officeDocument/2006/relationships"
4       xmlsns:w="http://schemas.openxmlformats.org/wordprocessingml/2006/main"
5       xmlns:w14="http://schemas.microsoft.com/office/word/2010/wordml"
6       xmlns:w15="http://schemas.microsoft.com/office/word/2012/wordml" ms:Ignorable="w14 w15">
7       <w:frameset>
8           <w:framesetSplitbar>
9               <w:w w:val="60"/>
10              <w:color w:val="auto"/>
11              <w:noBorder/>
12          </w:framesetSplitbar>
13          <w:frameser>
14              <w:frame>
15                  <w:name w:val="1"/>
16                  <w:sourceFileName r:id="rId1"/>
17                  <w:linkedToFile/>
18              </w:frame>
19          </w:frameser>
20      </w:frameset>
21      <w:optimizeForBrowser/>
22      <w:allowPNG/>
23  </webSettings>
```

**Figure 2.** The content of the "word\webSettings.xml" in a DOCX document.

### 4.2.2. Image Document Structure Path

After the image document passes through the document parser corresponding to the image, the order of appearance of each object in the document relative to the byte order of the document is obtained. According to this object order, the n-gram method is used to count the object structure paths with the most malicious images in the image document, and the term frequency–inverse document frequency (TF-IDF) algorithm is then used for feature selection.

### *4.3. Code Keywords*

Word documents often contain macros or VBA modules for automated office tasks. PDF documents often contain JavaScript code, which can support advanced functions such as form filling, URL requests, and starting external processes. Therefore, it is particularly important to correctly identify the security of the code embedded in the document.

Before embedding the JavaScript code, attackers generally obfuscate the code to increase the difficulty of analysis by security personnel. The readability of the code is reduced after obfuscation, and it is difficult to restore the obfuscated code to the original code. In particular, it is very difficult to extract the features of JavaScript code from PDF documents, which has been confirmed by previous work. Therefore, the extraction of code-related features from PDF documents was not considered in the present work.

The code obfuscation problem also exists in Office documents, and the code of the VBA program will increase the difficulty of analysis after obfuscation [49,50].

It is challenging to directly de-obfuscate the code; moreover, it will increase the analysis time, and the de-obfuscation effect will greatly affect the extraction of features in subsequent code. Therefore, research on the code itself should be avoided as much as possible during the document analysis.

### 4.3.1. Office Document Code Keywords

The keywords in the VBA code can indicate the behavior of the VBA code. The meanings of commonly used VBA keywords are provided in Table 3.

**Table 3.** The functions of some VBA keywords.

| Keywords | Description |
|---|---|
| VirtualAllocEx | May inject code into external processes |
| new object | May inject code into external processes |
| ScriptBlock | May run a PowerShell command |
| xxx.exe | May run an external program |
| Xor | May try to obfuscate special strings |
| FindWindow | May enumerate all windows |
| AutoOpen | Run code automatically when the document is opened |
| user-agent | May download files from the website |
| xxx.lnk | May execute malicious scripts to collect and upload sensitive files of users on the computer |
| base64 string | May have confused the string |
| vmware | May have confused the string |
| popen | May run an executable file or execute system commands on a MAC computer |

Oletools can be used to extract the VBA code in Office documents. Furthermore, it is also possible to directly extract code keywords appearing in Office documents without de-obfuscating and analyzing the VBA code. This can significantly reduce the document processing time while abandoning the de-obfuscating analysis of the VBA code. Moreover, although the keyword features of the obfuscated VBA code are hidden, obfuscation techniques are used in the code; thus, keywords such as XOR and Base64 String also exist in the code.

The keywords originally extracted from Oletools require further processing. Irrelevant keywords (such as "print") are filtered, and the keywords with URL and IP addresses are organized in a unified way to reduce the types of keywords. For example, the VBA codes containing IP addresses will be tagged with "ipv4," and the VBA codes containing "xx.exe" will be tagged with ".exe."

Aiming at the code keyword characteristics of the macro-virus of the XLS document, 19 code keywords distributed in the workbook and book documents after decompressing the XLS documents were manually extracted as candidate features. These keywords were as follows: Save It, Comma_laroux, Internat.exe, Application.StartupPath, Auto_Close, Classic.Poppy, Document_array, Auto_ouvrir, _VBA_PROJECT, Antivirus, Excel.Sheet, Normal_tabe, Tabelle2\x85\x00, Foglio1\x85\x00, Milliers results\x85\x00, Comma_Exec, Sayfa1\x85\x00, and LOMHNMKQKDY\x85\x00.

### 4.3.2. Image Code Keywords

The image parser correctly recognizes the data that do not meet the specification, but there remain data that do not meet the specification in benign images. This portion of the data also includes iframe tags, HTML codes, and JavaScript codes. This indicates that the appearance of code outside the specification does not necessarily increase the suspiciousness of the document, and the specific reason lies in the content of the code. Thus, representative keywords must be extracted from the code to characterize the degree of maliciousness of the code. The code processing includes three stages, namely code extraction, code preprocessing, and code n-gram analysis.

Code extraction: In this stage, the code characteristics of the bytecode of the data outside the specification are analyzed, and the bytecode range of possible code fragments is searched by traversal. The bytecode length of the code can be set to a minimum of 18 bytes, and the chardet module in Python can then be passed to detect the character string encoding method of the bytecode. Finally, the bytecode decode is converted into a character string to obtain the initial code.

Code preprocessing: The code will contain special symbols such as URLs, binary-coded strings starting with "#," and newline and tab characters. The operations include the following:

1. The characters are converted to lowercase;
2. The specific content of the URL is ignored;
3. The specific content of the binary code string is ignored; and
4. Special symbols, such as tabs and line breaks, are deleted.

The preprocessing of the code is completed via these four steps, and the code text is obtained after simplifying the volume and content.

Code n-gram analysis: The n-gram algorithm is applied to the code text obtained in the previous step to obtain code features within a certain fragment length range. In the later stage, relevant feature extraction algorithms can be used to extract the keywords of the key code as features.

### 4.4. Number of Objects

The number of objects in the document will also be significantly different, whether they are stream objects, OLE objects, or compressed objects. Maiorca et al. [25] obtained a high detection rate by training the number of various objects in a PDF document as a feature. It was found that there are significant differences between the numbers of objects in malicious and benign documents.

Based on this theory, in the proposed framework, the numbers of objects generated after document parsing are also trained as features.

#### 4.4.1. Number of PDF Document Objects

Ten object quantity characteristics were obtained from the data parsed by the PeePDF tool, including the following: (1) the size of the document, (2) the number of document versions, (3) the number of streams, (4) the number of compressed objects, (5) the number of object streams, (6) the number of X-ref streams, (7) the number of objects containing JavaScript, (8) the compression filter algorithm used, (9) the number of URLs, and (10) the algorithm used.

Although these features are weak for judging the maliciousness of documents when used alone, together they provide a good overview of the entire PDF structure. For example, in the authors' experience, the size of a malicious PDF document (and the number of objects or streams) is usually smaller than the size of a benign PDF document, and this statement is reasonable. A malicious PDF may not contain text, because the smaller the document size, the less time it takes to infect a new victim. Similarly, objects and X-ref streams are usually used to hide malicious objects in documents, while compressed objects can include embedded content, such as script code or other EXE/PDF documents.

#### 4.4.2. Number of Office Document Objects

In the Word document detection process, the type of OLE document, the number of malicious images, and the number of malicious URLs are used as features to enhance the detection rate of malicious Word documents.

(1) Number of OLE document types

It is troublesome to directly analyze an embedded OLE document. Based on the analysis of the types of OLE documents embedded in the samples, it was found that the benign samples contained only documents with bytecodes of all $0 \times 00$ and some binary documents with unknown document types. Numerous types of documents were found in the malicious samples, such as HTML, PE, DOS, JavaScript, and JAR documents.

This indicates that Word documents with OLE object embedding are likely to be embedded with programs containing malicious code or media documents containing vulnerabilities, while OLE objects similar to the executable programs are rarely seen in normal samples; thus, this will be beneficial for the detection of malicious documents.

Oletools is an open-source security analysis tool developed in the Python language for the analysis of OLE objects, VBA code extraction, and the VBA code analysis of Office documents. The OLE objects contained in each sample can be extracted and saved as

documents via the command line, after which Exiftool can be used to identify the type of the extracted document. The use of the OLE document type as a feature not only relies on the result of Exiftool, but also combines the document suffixes corresponding to the extracted OLE objects as features, as malicious documents usually disguise malicious OLE objects to reduce user suspicion.

(2)    Number of malicious images

Images usually exist in Word documents, and malicious images usually have potential vulnerabilities. Attackers often use Office tools to parse the potential vulnerabilities of images to launch attacks. Therefore, it is particularly important to analyze the security of images. The proposed static detection framework is used to detect the security of the images in a Word document after completing the image security test. For each document, a numerical value is used to record the number of malicious images in the document.

(3)    Number of malicious URLs

A malicious URL request will automatically download the malicious document and open the website. The content that can initiate a URL request in DOCX is mainly determined by the "Target" attribute of the "Relationship" tag in the ".rels" document, excluding the "document.xml.rels" document in the "word/_rels" directory of the ZIP document. If the value of "Target" is a URL link and the link ends with the document name (the document download link), the danger of the DOCX document is significantly enhanced.

Therefore, in the DOCX document detection process, the URL will be extracted from the ".rels" described in the previous paragraph, and the number of URLs will be used as a feature.

### 4.4.3. Number of Image Objects

Via the image document parser, the number of each type of object in the document is obtained after parsing the document, and the sum of the number of each type of object is used as a feature.

### 4.4.4. Number of RTF Objects

The content analysis of RTF documents is realized by writing an RTF document parser. The sequence and the number of occurrences of each type of object and each type of keyword in the document are obtained from the analysis result.

The RTF document contains many kinds of keywords, and the latter portion of the keywords usually contains numbers, which significantly increases the number of types of keywords. To avoid the problem of keyword expansion, the numbers in the latter part of the keywords are ignored when parsing RTF documents.

Some keywords are selected from the set of keywords that appear in the RTF documents, and the number of times that they appear in the document is used as a feature. Via the statistics of the samples, some substantial differences were found between the keywords of the malicious and benign samples.

In addition, the quantitative characteristics of OLE objects embedded in RTF documents are also analyzed. Here, Oletools is used to extract OLE objects embedded in RTF documents.

## 5. A Universal Static Detection Framework for Malicious Documents based on Feature Generalization

This paper proposes a universal static detection framework for malicious documents based on feature generalization. The framework mainly detects the maliciousness of documents from four aspects, namely specification check errors, the document object structure, document code analysis, and the number of document objects.

### 5.1. Composition of the Detection Framework

The universal static detection framework for malicious documents based on feature generalization is composed of document parsing, feature extraction, feature selection, parameter tuning, classifier, and detection report modules. The composition is presented in Figure 3.

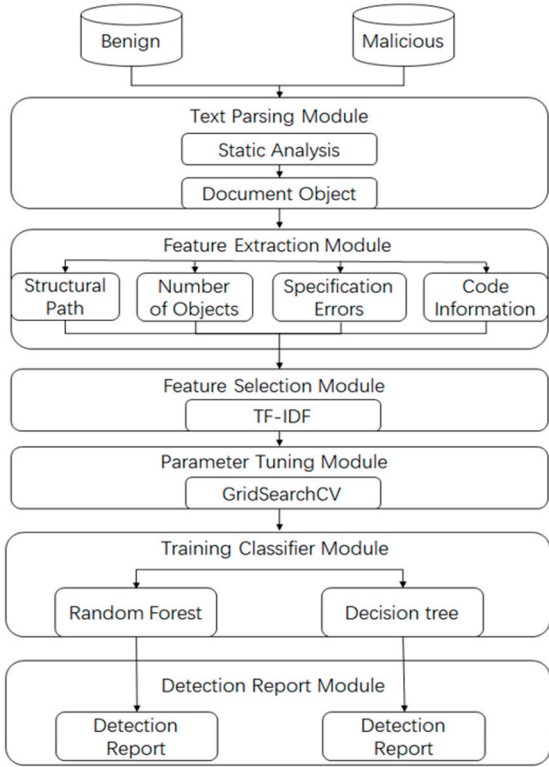

**Figure 3.** The framework of the proposed malicious document static detection system.

The document parsing module uses the corresponding static document parsing tool to parse the document into document objects according to the document type, and records error information and error locations during the parsing process.

The feature extraction module extracts the structure path information of the document, the quantity information of various objects, the specification document error information, and the information of the code embedded in the document according to the document analysis result.

The feature selection module uses feature extraction algorithms to extract the most effective features after obtaining various types of information about the document.

The parameter tuning module uses the GridSearchCV method to train the best parameters for the classifier.

The classifier uses artificial intelligence algorithms to make judgments on the maliciousness of documents. The pythonsklearn module is used to further develop the classifier. The classifier includes two classification algorithms, namely the decision tree and random forest algorithms, both of which are characterized by a fast speed and high accuracy. Moreover, they can both process high-dimensional data, and they are suitable for the static detection of malicious documents.

The detection report module generates detection reports.

### 5.2. Feature Selection Algorithm

In the feature selection method, the TF-IDF model is used to select features. This model was mainly used for information retrieval in the early days. If word $i$ has a high frequency of occurrence in article $j$ but low frequency in other articles, it means that word

*i* has a good ability to distinguish articles; therefore, it is suitable to use *i* to distinguish article *j* from other articles. TF-IDF can also be used for keyword extraction, which is used to extract features under this framework. The TF-IDF model is shown in Equation (1). For word *i* in document *j*:

$$W_{i,j} = tf_{i,j} \times \log\left(\frac{N}{df_i}\right) \tag{1}$$

where the $tf_{i,j}$ is the number of times the word *i* appears in the article *j*, $df_i$ is the number of words *i* in the article, and *N* is the total number of articles.

The feature selection process conducted in the experiments carried out in the present study was as follows. First, the features of the benign and malicious documents were extracted to generate the benign document candidate feature set and the malicious document candidate feature set, and the distribution number of each feature was then separately counted. Finally, feature selection was performed for each feature, and the ratio of the number of this feature in the benign samples to its number in the malicious samples was calculated. If the ratio was greater than the threshold, the feature was selected as the final feature.

## 6. Experiments and Results

### 6.1. Data Set

To collect extensive samples and maintain the representativeness and randomness of the distribution, malicious samples were collected from Contagio, VirusTotal, and VirusShare, and benign samples were mainly sourced from Google, Baidu Library, and academic sharing for academic purposes [48]. Ultimately, two datasets were obtained. Dataset1 included 290,542 files and was collected from Contagio, VirusTotal, VirusShare, Baidu, and academic sharing for academic purposes, while Dataset2 included 2918 files and was collected from Contagio and Google. Dataset1 and Dataset2 were collected at different times from different sources.

Contagio is a well-known blog that collects malicious samples for observation, analysis, and threat observation. VirusShare is an online sample submission website and has the function of sharing samples among members of the site. Each sample uploaded to the VirusShare website is monitored online on the VirusTotal website. Each sample is detected online by using more than 70 antivirus scanners to jointly detect a sample to evaluate the degree of maliciousness. If two or more antivirus programs detect a virus in the detection results of a sample, it is judged as a malicious sample; otherwise, it is judged as a benign sample.

The distributions of the two datasets are respectively presented in Tables 4 and 5. In addition, malicious document attacks usually do not use PPTX and Excel documents as the source of the attack. Therefore, malicious detection experiments were not conducted on these document types.

**Table 4.** The distribution of Dataset1.

| Type | Total | Benign | Malicious |
|---|---|---|---|
| PDF | 114,142 | 29,454 | 84,688 |
| DOC/DOCX | 49,066 | 15,341 | 33,645 |
| XLS/XLSX | 11,503 | 4430 | 7073 |
| GIF | 67,388 | 17,168 | 50,220 |
| JPEG | 33,825 | 18,585 | 15,240 |
| PNG | 7943 | 3983 | 3960 |
| RTF | 6675 | 3357 | 3318 |

**Table 5.** The distribution of Dataset2.

| Type | Total | Benign | Malicious |
|------|-------|--------|-----------|
| PDF | 500 | 250 | 250 |
| DOC/DOCX | 200 | 100 | 100 |
| XLS/XLSX | 400 | 200 | 200 |
| GIF | 250 | 200 | 50 |
| JPEG | 606 | 300 | 306 |
| PNG | 562 | 262 | 300 |
| RTF | 400 | 200 | 200 |

### 6.2. Experimental Environment and Parameters

This experiment was carried out on the Windows 10 platform, the Python version was 3.5.2, and the hardware parameters of the machine were an Intel® Core™ i7-8700 CPU @ 3.2 GHz with 16 GB RAM.

Because the decision tree and random forest models are characterized by the advantages of being able to process large data sets, balancing the error of unbalanced data sets, a fast training speed, and good performance, these models were ultimately chosen for use in the experiments. For the decision tree classifier, the default parameters provided by Sklearn were used to modify criterion = "entropy."

The parameters of the random forest were obtained by automatic tuning via the use of the GridSearchCV method provided by Sklearn; GridSearchCV generates all candidate parameters from the grid parameter values determined by the parameter (param_grid), and finally selects the set of parameters with the highest score.

The default parameters of the decision tree and random forest models are reported in Table 6.

**Table 6.** The default parameters of decision tree and random forest models.

| Decision Tree | Random Forest |
|---------------|---------------|
| Criterion = "gini" | n_estimators = 100 |
| Splitter = "best" | Criterion = "gini" |
| max_depth = None | max_depth = None |
| min_samples_split = 2 | min_samples_split = 2 |
| min_samples_lea f = 1 | min_samples_leaf = 1 |
| min_weight_fraction_leaf = 0.0 | min_weight_fraction_leaf = 0.0 |
| max_features = None | max_features = "auto" |
| random_state = None | max_leaf_nodes = None |
| max_leaf_nodes = None | min_impurity_decrease = 0.0 |
| min_impurity_decrease = 0.0 | min_impurity_split = None |
| min_impurity_split = None | Bootstrap = True |
| class_weight = None | oob_score = False |
| Presort = 'deprecated' | n_jobs = None |
| ccp_alpha = 0.0 | random_state = None |
|  | Verbose = 0 |
|  | warm_start = False |
|  | class_weight = None |
|  | ccp_alpha = 0.0 |
|  | max_samples = None |

### 6.3. Experimental Results

6.3.1. Experiment based on Feature Generalization

The experimental dataset was divided into a training set (about 80%) and a test set (about 20%), and ten-fold cross-validation was conducted to obtain the average accuracy on the test dataset. The metrics for dichotomous problems are represented by the detection rate (TPR), the false positive rate (FPR), the accuracy rate (ACC), and the F1-score. The experimental results of various documents are reported in Table 7.

**Table 7.** The experimental results of various documents under the proposed static detection framework for malicious documents based on feature generalization.

| Type | Method | ACC (%) | TPR (%) | FPR (%) |
|------|--------|---------|---------|---------|
| PDF | Decision Tree | 99.11 | 99.43 | 1.83 |
|  | Random Forest | 99.37 | 99.40 | 0.78 |
| DOC | Decision Tree | 97.16 | 97.11 | 2.40 |
|  | Random Forest | 97.40 | 97.23 | 0.82 |
| DOCX | Decision Tree | 98.28 | 95.71 | 0.67 |
|  | Random Forest | 99.04 | 97.36 | 0.27 |
| XLS | Decision Tree | 93.89 | 92.91 | 2.12 |
|  | Random Forest | 94.66 | 93.64 | 1.17 |
| GIF | Decision Tree | 99.50 | 99.58 | 0.73 |
|  | Random Forest | 99.88 | 99.89 | 0.17 |
| JPEG | Decision Tree | 96.57 | 94.12 | 1.39 |
|  | Random Forest | 96.76 | 94.54 | 1.34 |
| PNG | Decision Tree | 98.63 | 98.63 | 1.37 |
|  | Random Forest | 98.50 | 97.64 | 0.615 |
| RTF | Decision Tree | 97.81 | 95.93 | 0.43 |
|  | Random Forest | 98.68 | 97.59 | 0.28 |

Based on the experimental results, it is evident that the proposed static detection framework for malicious documents based on feature generalization achieved a higher TPR and a lower FPR in the malicious detection experiments of various documents. The TPR of malicious samples and the FPR of benign samples of the classifier obtained by the random forest were found to be better than those of the classifier obtained by the decision tree. It was only in the detection of malicious PNG documents that the detection rate declined. The overall performance of benign sample detection was good, but the FPRs of JPEG and XLS documents were somewhat high.

In the subsequent experiment, the random forest was used as the classifier. Table 8 reports the detection accuracies of different types of files on Dataset2. The experimental results verify that the proposed method exhibited good scalability performance.

**Table 8.** The detection accuracies of different documents.

| Type | ACC (%) | TPR (%) | FPR (%) | F1-Score |
|------|---------|---------|---------|----------|
| PDF | 99.2 | 98.4 | 0 | 0.9919 |
| DOC/DOCX | 98 | 99 | 0.03 | 0.9802 |
| XLS/XLSX | 97.25 | 97 | 0.025 | 0.9724 |
| GIF | 100 | 100 | 0 | 1 |
| JPEG | 97.69 | 95.42 | 0 | 0.9766 |
| PNG | 99.47 | 99 | 0 | 0.995 |
| RTF | 99.25 | 98.5 | 0 | 0.9924 |

6.3.2. Comparison with Antivirus Software

The proposed method was compared with commonly used antivirus software, including Kaspersky, NOD32, and McAfee. The virus database was updated in October 2021.

As shown in Table 9, compared with Kaspersky, NOD32, and McAfee, the proposed static detection framework for malicious documents achieved superior accuracy and recall rates for the detection of various types of documents; however, the FPR of benign samples was slightly increased. The main reasons for this are as follows:

1.  To reduce the detection time, the number of features selected in the experiment was small. For example, seven important features can be selected as regular features for

PDF document detection [10], but only four important features were selected in this experiment;

2.  The antivirus software has a huge virus database, and the sample size of this system is not sufficiently large; this led to a slight increase in the FPR.

**Table 9.** The experimental results of the detection of different types of malicious documents by different antivirus software.

| Type | Software | ACC (%) | TPR (%) | FPR (%) | F1-Score |
|------|----------|---------|---------|---------|----------|
| PDF | Kaspersky | 97.47 | 96.59 | 0.0102 | 0.9827 |
|  | McAfee | 89.92 | 86.43 | 0.0407 | 0.9271 |
|  | NOD32 | 95.22 | 93.57 | 0.0204 | 0.9667 |
|  | Ours | 99.37 | 99.40 | 0.78 | 0.9956 |
| DOC | Kaspersky | 46.17 | 26.74 | 0 | 0.4219 |
|  | McAfee | 84.46 | 78.86 | 0.0086 | 0.8818 |
|  | NOD32 | 55.54 | 39.51 | 0.0344 | 0.5663 |
|  | Ours | 97.40 | 97.23 | 0.82 | 0.9845 |
| DOCX | Kaspersky | 94.33 | 80.45 | 0 | 0.8917 |
|  | McAfee | 96.30 | 87.25 | 0 | 0.9319 |
|  | NOD32 | 96.44 | 87.85 | 0.0540 | 0.9346 |
|  | Ours | 99.40 | 97.36 | 0.27 | 0.9834 |
| XLS | Kaspersky | 42.03 | 6.06 | 0 | 0.1143 |
|  | McAfee | 86.63 | 78.33 | 0 | 0.8785 |
|  | NOD32 | 48.81 | 17.05 | 0 | 0.2913 |
|  | Ours | 94.66 | 93.64 | 1.17 | 0.9635 |
| GIF | Kaspersky | 53.74 | 37.93 | 0.0349 | 0.5500 |
|  | McAfee | 27.13 | 2.23 | 0.0233 | 0.0436 |
|  | NOD32 | 26.34 | 1.15 | 0.0058 | 0.0228 |
|  | Ours | 99.88 | 99.89 | 0.17 | 0.9992 |
| JPEG | Kaspersky | 76.50 | 49.10 | 0.0107 | 0.6586 |
|  | McAfee | 78.40 | 53.33 | 0.1017 | 0.6951 |
|  | NOD32 | 58.99 | 11.15 | 0 | 0.2007 |
|  | Ours | 96.76 | 94.54 | 1.34 | 0.9638 |
| PNG | Kaspersky | 81.60 | 63.32 | 0 | 0.7754 |
|  | McAfee | 50.48 | 1.27 | 0 | 0.0250 |
|  | NOD32 | 49.86 | 0.02 | 0 | 0.0005 |
|  | Ours | 98.50 | 97.64 | 0.615 | 0.9850 |
| RTF | Kaspersky | 99.28 | 98.64 | 0.0894 | 0.9927 |
|  | McAfee | 94.83 | 89.63 | 0.0298 | 0.9452 |
|  | NOD32 | 96.30 | 92.59 | 0.0298 | 0.9614 |
|  | Ours | 98.68 | 97.59 | 0.28 | 0.9864 |

As shown in Table 9, the FPRs of Kaspersky, NOD32, and McAfee may have been zero. The main reasons for this situation are as follows:

1.  If the antivirus software mistakenly reports a benign document as a malicious document, this will cause serious consequences; the antivirus software will try to reduce the FPR;

2.  Antivirus software may not be perfect for this type of document detection. Although the FPR was 0, the ACC was also low;

3.  The number of samples in the comparison experiment may not have been sufficient, and there were no samples that could have caused the antivirus software to falsely report; and

4.  Some types of documents have fewer vulnerabilities that can be used for virus embedding, and antivirus software is characterized by more complete detection for this type of document; thus, there also occurred the situation in which the FPR was 0 and the accuracy rate was also high.

As shown in Figure 4, in most cases, the F1-score of the proposed static detection framework for malicious documents was higher than those of the compared antivirus software. This proves that the proposed framework achieved good performance in the detection of various types of documents.

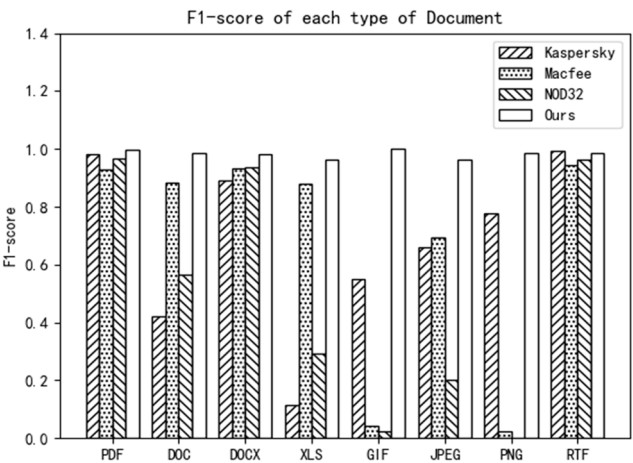

**Figure 4.** The F1-scores of various types of documents.

6.3.3. Comparison of Detection Time

Table 10 reports the average detection times of different antivirus software and the proposed system. The average detection time is the average detection time for all types of files. It can be seen from Table 10 that the average detection time of the proposed method was very close to those of Kaspersky and McAfee. The average detection time of the proposed method even exceeded that of the Kaspersky software. Although the average detection time of NOD32 was shorter than that of the proposed method, the F1-score of the proposed method was higher than those of all three compared antivirus software. These time gaps are acceptable in reality.

**Table 10.** The average detection times of different antivirus software.

| Method | Average Detection Time (s) | Average F1-Score |
|---|---|---|
| Kaspersky | 0.7214 | 0.7462 |
| McAfee | 0.4887 | 0.743 |
| NOD32 | 0.152 | 0.6662 |
| Our method | 0.5926 | 0.9902 |

## 7. Discussion

Through the analysis of different types of samples in the experimental results, the proposed static detection framework for malicious documents based on feature generalization was found to have the following shortcomings.

(1) Some dynamic behavior characteristics appear only when the document reader software opens the document. It is difficult for the proposed method to obtain these behavioral characteristics of documents. The further analysis of undetected malicious PDF samples revealed that these samples had dynamic behaviors, including autorun, autoform, and autoaction;

(2) Obtaining only the number of OLE objects in Word documents cannot lead to the effective classification of the maliciousness of word documents; thus, additional tools are needed to further analyze the maliciousness of OLE objects;

(3) When judging the maliciousness of documents with only code embedded, the selected code keywords must be optimized to improve the accuracy of the code keywords. When using JavaScript code to detect samples, the detection rate of the proposed method will be reduced due to the low keyword coverage.

In practical applications, the proposed method can be combined with dynamic detection. After suspicious documents are found, dynamic detection methods can be used for further detection. Moreover, the code keywords extracted from the document require further optimization. Finally, additional tools can be used to further analyze OLE objects and discover more effective features for the judgment of maliciousness.

## 8. Conclusions

Researchers rarely study common types of documents other than PDF and Word documents. The present research compensates for this deficiency. In this research, the detection features of a certain type of document were generalized so that the features could be used to detect other types of documents. Moreover, a static detection framework based on feature generalization was proposed. The framework uses four feature dimensions, namely specification check errors, the (object) structure path, code keywords, and the number of objects.

The proposed method was verified on two datasets, and was compared with Kaspersky, NOD32, and McAfee antivirus software. The F1-score of malicious PDF detection of the proposed method was 0.9956, that of malicious DOC detection was 0.9845, and that of DOCX detection was 0.9834. Therefore, good detection rates were achieved for the two main document types of PDF and Word, and the FPR was within 1%. In addition, the proposed framework was found to achieve better results on other types of documents; the average F1-score of all these types of documents was 0.9902. The results of experiments demonstrated that the proposed framework achieved better detection performance than Kaspersky, NOD32, and McAfee antivirus software. Moreover, the average document detection time of the proposed method was found to be 0.5926 s, which was very close to that of McAfee (0.4887 s), and shorter than that of Kaspersky. The experimental results verify that the proposed method achieved good performance in terms of the detection accuracy, runtime, and scalability.

However, the proposed method is also characterized by some shortcomings. Some dynamic behavior characteristics appear only when the document reader software opens the document, and it is difficult for the proposed method to obtain these behavioral characteristics of documents. Moreover, the sole reliance on the number of OLE objects in the Word document is not sufficient. The selected code keywords must be optimized to improve the accuracy of code keywords.

In the future, the proposed method can be combined with dynamic detection. After suspicious documents are found, dynamic detection methods can be used for further detection. Moreover, additional tools can be used to find more effective features, and some features need to be further optimized as well.

**Author Contributions:** Conceptualization, X.L. and F.W.; methodology, X.L. and F.W.; software, F.W.; validation, X.L., F.W. and C.J.; formal analysis, X.L. and F.W.; investigation, X.L. and F.W.; resources, F.W.; data curation, F.W. and C.J.; writing—original draft preparation, X.L. and F.W.; writing—review and editing, X.L. and C.J.; visualization, F.W. and C.J.; supervision, X.L. and P.L.; project administration, X.L., F.W. and C.J. All authors have read and agreed to the published version of the manuscript.

**Funding:** This research was funded by National Natural Science Foundation of China (Grant No. 62136006), and the National Key R&D Program of China (Grant No. 2020YFB2104700).

**Institutional Review Board Statement:** Not applicable.

**Informed Consent Statement:** Not applicable.

**Data Availability Statement:** Not applicable.

**Conflicts of Interest:** The authors declare no conflict of interest.

## Appendix A. Malicious Forms of Image Files

*Appendix A.1. Forging File Header*

Forging file header is to compose a file using the file header logo of the image file and malicious code, and save the file in a specified format. The most commonly used forging method is to store a piece of Personal Home Page (PHP) code behind the GIF file header [42]. Its purpose is to attack an insecure Web site that only checks the uploaded file header (also known as the magic number) and ignores the legality check of the file content, which result in a file with code being uploaded to the site. Then the attacker accesses the uploaded file to make the code run. A specific method is to add the magic number "GIF89a" in front of a PHP file (the magic number is the file header of the GIF file), and modify the file suffix name to GIF. Then the attacker starts subsequent operations by uploading the file through the file upload function of the website. Figure A1 is a text display result of a malicious GIF sample.

```
GIF89a?lovealihack <?php
eval($ POST[alihackxx])?>
```

**Figure A1.** A sample of a PHP file that added a GIF file header.

*Appendix A.2. CVE Vulnerability*

There are many image analysis and editing tools on the market, such as Office software and ImageMagick tools. These tools used to have some vulnerability in processing images. Taking the Office software as an example, the attacker uses the abnormal data in the JPEG file to trigger the software vulnerability, and then executes the shellcode embedded in the JPEG file to finally realize the attack [43].

*Appendix A.3. Insert Code*

Commonly, the codes inserted in picture files include JavaScript code, HTML code, PE file, PE file import table (IAT), etc. [44]. JavaScript codes are most commonly found in JPEG pictures, GIF and PNG pictures. HTML codes mainly contain some iframe tag. HTML webpage codes can be attached to the end of picture files. In addition, a PE file and the import table of the PE file also can be attached to the end of picture files.

*Appendix A.4. Malicious Data Hiding*

In a GIF file, there is a data segment of the comment extension area, and the picture creator can add additional information in the comment extension area. Most image analysis software generally skip the comment extension area directly when analyzing images, so this area is rarely noticed. Some malicious pictures include some additional information in this area, such as URL and encoded data. For JPEG pictures, people can use steganography to embed malicious code or malicious data in certain areas of the file [45].

In addition to adding malicious data to the internal structure of a picture, the attacker will also add other file data after the picture data to deceive users to pass the malicious file. These file types include PE files, RAR compressed files, PDF files, DOC files, PNG pictures, JPEG pictures, GIF pictures, text, and other types of files.

## Appendix B. Malicious Forms of RTF Documents

The forms of RTF attacks mainly include array overflow vulnerabilities and OLE object vulnerabilities.

*Appendix B.1. Array Overflow Vulnerability*

The essential reason of the array overflow vulnerability is caused by the security of virtual functions in the virtual function table of polymorphic functions in C ++ programming [46]. The memory distribution of objects in C ++ is as shown in Figure A2. Since the objects in memory are distributed in order, the memory of the member object of the

above object is immediately followed by the virtual function table pointer of the next object. When the data member of the above object writes data and generates an overflow, the content of the virtual function table pointer of the next object will be overwritten. Then, when the virtual function is called, the wrong virtual function table is used, causing the program execution flow to change. When RTF documents are parsed, structures such as catalog tables may encounter array overflow problems.

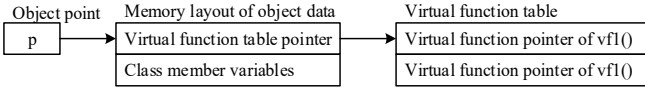

**Figure A2.** The general memory model of objects containing virtual table classes in C ++.

*Appendix B.2. OLE Object Vulnerability*

There are many control words in a RTF document, and different types of OLE objects can be embedded through different control words. For example, the \objemb control word can embed other documents, audio files, pictures and other information into RTF documents.

RTF documents can contain a variety of OLE objects, and the embedded OLE objects are parsed when the document are opened. Various vulnerabilities might be triggered when these OLE objects are parsed [47]. The exploit can be used to execute malicious code. For example, when the OLE object of the TreeView control is embedded in a RTF file, this control causes a stack overflow problem when parsing the data in the \objdata section. This vulnerability can be used to trigger problems such as remote code execution. Parsing the OLE object of the embedded TabStrip control can cause memory corruption.

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
