# Peer review of "A Universal Malicious Documents Static Detection Framework Based on Feature Generalization"

_applsci, doi:10.3390/app112412134_

Round 1

Reviewer 1 Report

The authors propose a universal static detection framework for malicious documents based on feature generalization. 
The framework mainly detects the maliciousness of documents from various aspects, i.e., specification check errors, 
document object structure, document code analysis, and number of document objects.

The paper should be shortened. 
Unproven claims should be checked, especially in Section 3, see below.

95: Especially, Section 2 about related work has many similarities to a paper that two of the authors had published in 2019.
Xiaofeng Lu, Fei Wang, Zifeng Shu. "Malicious Word Document Detection Based on MultiView Features Learning" 28th International Conference on Computer Communication and Networks (ICCCN), 2019.
I consider not referencing this paper to be significant.

235: Section 3 "Research on the Attack Forms of Malicious Documents" does not contain any references.
This is surprising as I doubt that nobody else has done research on this topic.
(Whouldn't "Attack Forms of Malicious Documents" be a better title of Section 3?)

259: are these the only two methods?
272: where do these five categories come from?
282: where do these aspects come from?
295: how do you know?

298: You write that through the analysis of several malicious documents, you conclude several attack methods 
of malicious documents. How many are several? How did you come to your conclusions?

325: Is the Offvis tool still supported by Microsoft?

337,363,365: Figures 1,2,3: are they really necessary? Do they have to be that big?

370,406: How did you get the data shown in Tables 1 and 2? You did not specify the source yet.

388: You 'extracted 4 error-related information as features'. What was the condition that sth. became a feature?

582: Are there any facts that support your statement? Or is it pure assumption?

Reviewer 2 Report

The paper present an approach to the malicious document static detection, which is based on feature generalization. The addressed features include the specification check error, structure path, code keywords and number of objects.

The authors deal with an important problem. Overall, the paper is well-organized and fluently readable. The proposed solution is interesting. The empirical evidence appears to be promising. However, there are some issues, which are discussed below.

The Introduction must be refined. The authors have to clear exemplify the type of threat, which can be associated with each of the PDF, Word, Excel, RTF, JPEG, PNG and GIF document types. Possibly, the discussion concerning the PDF and Word types can be dealt with in the introduction, while all other document types can be discussed in the appendix. I believe that this is going to be a significant contribution of practical interest, if the authors presents examples drawn from real-world cases. Also, the novelty behind each contribution has to be further stressed in the Introduction. Yet, the plan of the manuscript has to be added to the Introduction.

Section 3 has to be more thoroughly justified as separate from Section 2.

Without prejudice to the usefulness of the approach on a purely application level, the novelty of the approach on a scientific level is not high.

The empirical validation is performed on only one dataset, which does not appear to be large-scale.

The runtime and scalability of the proposed approach have to be extensively studied in a comparative fashion.

Section 7 has be revised to more insightfully discuss the disadvantages of the proposed approach, together with their impact on practical applications.

Finally, the Conclusions section has to be refined to reflect the evidence from comparative runtime and scalability analysis as well as the deepened discussion on the disadvantages of the proposed approach.

Round 2

Reviewer 2 Report

The paper was improved and it is now suitable for publication.

However, the authors must make sure that their answer to my first question is fully reported in the Introduction as it appears in their cover letter. Actually, I noted some discrepancy.

Moreover, in the caption of Table 7 and 8, the name of the tested datasets should be reported.